# Healthy Leadership and Workplace Health Promotion as a Pre-Requisite for Organizational Health

**DOI:** 10.3390/ijerph18179260

**Published:** 2021-09-02

**Authors:** Isabell Koinig, Sandra Diehl

**Affiliations:** Department of Media and Communications, University of Klagenfurt, 9400 Klagenfurt, Austria; Sandra.Diehl@aau.at

**Keywords:** organizational health, healthy leadership, workplace health promotion, qualitative study

## Abstract

(1) Background: Increasing stress levels at the workplace constitute a concerning organizational trend, challenging not only employees but also organizations alike, as it is in most instances associated with increasing workloads. In consequence, employees have started to demand that organizations begin to accept responsibility for their health and well-being. The present contribution seeks to investigate, to which extent individuals are able to deal with stress and whether their employers and respective supervisors (leaders) accept responsibility for their health, for instance, by leading by example. In addition, the existence and support generated by the organization in form of Workplace Health Promotion (WPHP) is inquired. (2) Methods: Semi-structured qualitative interviews with 40 (full and part-time) employees from two European countries were conducted. (3) Results: The study with employees from Austria and Germany (*n* = 40) confirmed that employees have started to recognize the potential of the workplace as an environment, where individual health can be enhanced. Yet, the results showed that only a few companies have already put some WPHP measures into practice. Likewise, the implementation of healthy leadership is rather limited to date. (4) Conclusions: At present, companies are still more likely to delegate responsibility for employee health and well-being to their staff, having not fully realized the potential of healthy leadership and organizational health promotion. There is great potential to increase WPHP measures on the employer side, through both healthy leadership and supporting WPHP measures.

## 1. Introduction

Increasing stress levels at the workplace constitute a concerning organizational trend [1,2,3,4]. The 2020 Trends Report on Well-Being at Work concluded that work is taking its toll on people and their health, respectively [5]. For some years already, work has been listed amongst the top three sources of stress [6]. In most instances, work-related stress was found to also have effects on individuals’ health, leading to sleep deprivation or mental health issues amongst others [3,7] and negatively affecting employees’ private lives [1,2]. In consequence, the Stress Awareness Month 2020 was dedicated to identifying and alleviating Work-Related Stress and Anxiety [8]. In most instances, employees’ stress stems from the high workloads they have to deal with on a regular basis [9]. Present-day work demands have found that 40% of people work more than 50 hours per week; an additional 74% claims to be sleep-deprived [10]. As this work overload contradicts the so-called “healthy job”—defined as the demands on employees being appropriate in relation to their abilities and resources [11]—it comes as no surprise that 43% of US employees think their employers should also be concerned with their health [12].

With the literature excessively documenting the detrimental aspects of employment, such as stress and burnout [13], organizations have been increasingly encouraged to show concern for their employees’ health and well-being. A growing concern for individual health in the workplace is grounded in positive organizational behavior, defined as “the study and application of positively-oriented human resource strengths and psychological capacities that can be measured, developed, and effectively managed for performance improvement in today’s workplace” ([14], p. 57). This approach induces organizations not only to change their behaviors [15] but also recognizes the relevance of human capital as an organization’s most valuable asset in an ever-changing environment [14]. Previous research has addressed the benefits of healthy workers for companies, such as higher job satisfaction, higher employee engagement, and better job performance [16,17,18,19,20,21,22], together with healthy leadership [23,24], which can be defined as the “health-specific influence of leaders on employee health and wellbeing” ([25], p. 2). In response to social movements and global initiatives concerned with improving health and well-being on a population level (e.g., see Sustainable Development Goals (SDGs) by the United Nations [26]; see also, Reference [27]), organizations have started to allocate resources towards individual health [28,29,30] in the form of Workplace Health Promotion (WPHP) measures. Especially against the background of a growing mediatization, which leads to blurring boundaries between employees’ private and professional lives and to an always-on mentality among many employees, individual health is repeatedly put to the test [31]. In consequence, the maintenance of a sound work-life-balance is becoming more and more important, as is the introduction of flexible work solutions, such as the ability to work from the home office, to work flexible hours during the day, or to access the company email account from home, which can improve individual well-being in the long run. While companies make more efforts related to their employees’ health, each individual also has responsibility for his/her own health. Healthy leadership can only be successful if both employers and employees work in concert to improve employee health and create a healthy work environment. However, the roles of who bears more responsibility are not yet clearly defined. In our study, we want to analyze whether employees perceive to have the responsibility for their health mainly themselves or whether they see their employers as responsible for their health as well. To date, this aspect is absent from research to the largest part.

Changes in Human Resources’ orientation are, first and foremost, conditioned by a paradigm shift, which stresses the necessity to invest in competitiveness (and skilled employees) instead of boosting productivity [32,33,34]. Since human resources are hard to copy, they constitute an organizations’ rarest and most valuable resource [35] and are part of the organization’s movement towards sustainability [36]. In the current “war for talent”, where companies are trying to get and keep qualified and capable employees, it is increasingly important that employers make employees feel that they care about them and that their health is not just a personal matter but also something that the company pays attention to.

In the following, focusing on an employees’ perspective, we aim (1) to analyze whether employees feel mainly responsible for their own health or whether they see the responsibility for their health to also lie with their employer and (2) to investigate the strategies employees use to establish a sound work-family-balance. While these are the main goals of our study, we also want (3) to explore whether, from an employees’ perspective, companies have already started to implement healthy leadership and (4) to determine which WPHP measures have already been implemented by companies, on the one hand, and how they are perceived by employees, on the other hand.

## 2. Theoretical Background

### 2.1. Organizational Health

Occupational health as a subject area was first introduced in the 1990s and accounted for the valuable contribution of psychologists to creating a healthy workplace [37]. In general, occupational health focuses on the creation of “healthy workplaces in which people may produce, serve, grow, and be valued, [and in which they] use their talents and gifts to achieve high performance, high satisfaction, and well-being” ([38], p. 3). While the term well-being is not easy to define, it usually refers to an individuals’ optimal functioning, as well as personal growth and development [39,40]. These concerns for individual well-being and health arose out of increasing psychosocial hazards individuals were confronted with on a daily basis [41], such as increasing work demands and poor work–life balance [42] or even work–life conflict [43,44], which impeded their work’s success and health. An overview over occupational health research themes showed that work-related stress and work–life balance were the most commonly researched areas [42]. Work–family balance—which is often used interchangeably with work–life balance—can be defined as “satisfaction and good functioning at work and at home with a minimum amount of role conflict” ([45], p. 349). Responsibility for health can be assigned to the management or be extended to an organization’s employees [46,47]. The first aspect recognizes the organization as a place where individual health can be improved. In this context, the workplace is seen as an environment “for promoting and maintaining improved levels of health over time” ([48], p. 141). In the second instance, employers are seen to be in the position to assist their employees in successfully juggling the different domains of their lives without any difficulties and tensions. Both forms of health support have been found to benefit the organization in the long run [49,50,51]. Thus, both the individual and the employer are responsible for enabling and maintaining a good work–family balance. To date, little research has been done on who is more responsible for employee health, the employee or the employer. This is a research gap that our study intends to fill.

In recent years, the term occupational health has been substituted by organizational health [52]. Bauer and Jenny [53] offer the following reasons for this shift: (1) employee health is more and more subject to the organizational context, (2) employee health has implications for the organization as a whole, and (3) both concepts are reciprocally linked and continue to influence one another. They offer a generic organizational health framework that is made up of the following elements: employees, employees in leadership positions, work processes, social processes, and the organizational environment [52,53]. Stress has been identified as an environmental cause of ill health [54], which can be overcome if the organizational parameters were changed. This assumption is supported by the person–environment fit model [55], which perceives the environment as essential in shaping individual responses to work situations. Thereby, the fit between the person (and its needs respectively) with the environment is seen as crucial [56]. Constituting an “ultimate competitive advantage” [57], organizational health describes an organization’s ability to operate effectively, grow sustainably, and adapt smoothly to change [58]. In addition, it is conceptualized as an organization’s ability “to align, execute, and renew itself faster than the competition to sustain exceptional performance over time. It comprises core organizational skills and capabilities, such as leadership, coordination, or external orientation, that traditional metrics don’t capture” ([57], p. 3).

### 2.2. Criteria and Strategies of Healthy Organizations

Organizational health comprises a number of areas, such as a company’s psychological and financial health. According to Cacace et al. [59], psychologically and financially healthy organizations are characterized as follows:(1)Healthy organizations set a clear direction for their future and how they conduct themselves.(2)Healthy organizations are executed well and have a culture of high performance.(3)Healthy organizations create a strong connection between employees and the company by showing appreciation and by bringing meaning to work.

These criteria correspond with two recent industry studies, according to which a shift in employee preferences can be observed: “What employees really want isn’t more money but better benefits. They want to work at a place that is really organizationally healthy” [60,61]. This suggests that individuals’ health and well-being is subject to their experiences of the workplace, as well as work-related aspects [62]. A healthy organizational environment consists of psychosocial factors that can enhance rather than harm individual health [63,64] and is expected to benefit the organization as a whole [64]. A healthy psychosocial environment, and encompasses all (environmental) social structures that have a bearing on individual health, such as work demands, control, and social support [65,66]. Moreover, workplace health and wellness programs are a common employee benefits [67] and describe the employer-provided efforts to “enhance awareness, change behavior, and create environments that support good health practices” ([68], p. 297). Employers believe that these programs reduce medical spending and increase the productivity [69].

The literature has defined a number of strategies and factors that influence organizational health and suggests that
(1)a balance between job demands, resources, job design, social relationships, and support, as well as change, need to be achieved [70,71];(2)organizational offerings need to be tailored to employee needs [72]; and(3)organizational health must be reflective of organizational climate. Thereby, organizational climate encompasses leadership and management practices, as well as organizational structures and processes [64,73].

Strategies to enhance organizational health mainly concern healthy leadership [74,75] and workplace health promotion [76]. Both aspects will be discussed in more detail in the following sections.

### 2.3. Healthy Leadership

Organizational health is linked to the concept of organizational politics [77], and as such, it has also increasingly been linked to positive leadership [14,74,75]. Leadership is conceptualized as a process in the course of which a person influences and directs others in an attempt to accomplish certain objectives or common goals [78]. Leaders have to ensure that organizations develop [79] and can motivate employees to follow their example [80,81]. As workplace health requires far-reaching organizational adaptations and changes, the management assumes responsibility for the process [82].

According to Hänsel & Kaz and Boehm et al. [23,24], healthy leadership is a new promising management approach, whereby an employer raises awareness for the topics of health and well-being at work. Following Rudolph, Murphy & Zacher [25], healthy leadership can be defined as the “health-specific influence of leaders on employee health and wellbeing”, whereby—as it is the case within the healthy leadership literature—health and well-being are understood to encompass physical, mental, and social well-being and not just the absence of disease [83]. Similar constructs are health-specific [84], health-focused [85], health-oriented [86], and health-promoting leadership [87,88] (for an overview of the different concepts, please see the review by Reference [25]). The objectives of a healthy leadership are to build trust, manage problems, and reduce the work-related pressure employees face [89]; healthy leaders are, furthermore, concerned with protecting, enhancing, and restoring the health of their employees [24]. The healthy leadership concept is based on two elements: (1) The employer him-/herself must live a healthy life ([23], p. 2); since the leader of a company often has high responsibilities, which results in a lot of stress and anxieties, a leader always sets an example for the employees and serves as a role model [90]. Applying the motivational theory of role modeling to the organizational health context, managers can serve as role models and, thus, have the ability to motivate individuals in both setting and achieving goals [91]. (2) Employers should implement Workplace Health Promotion measures to develop a health-promoting workplace that motivates employees to participate in such a development [92], which will be discussed in the next section and the empirical study.

### 2.4. Workplace Health Promotion

Increasingly, the workplace is recognized as a place in which employees’ health can be improved [93]. The concept of workplace health describes “the ability of the workforce to participate and be productive in a sustainable and meaningful way” ([82], p. 79). This increasing interest in workplace health is reflective of a growing concern for and interest in employee health and well-being as a means to reduce concerning retention rates [94,95]. In occupational health psychology, the term Workplace Health Promotion (WPHP) has been increasingly used [96,97]. As an organizational strategy, WPHP is part of the broader concept of organizational health [57].

Following the Practices for the Achievement of Total Health (PATH)-Model, healthy workplace practices can result in both employee and organizational health; thereby, both components are dependent upon each other [76]. A similar reasoning is offered by the heuristics model of occupational health, which postulates that individual and organizational behaviors can benefit employee well-being and organizational performance [98]. The health development model emphasizes that health and well-being are continuously produced and reproduced through interactions with individuals’ immediate environments [99]. All models are based on the assumption that health—a process that can be influenced by both the individual and the organization—can be maintained through concrete health-enhancing measures.

As part of a modern change management that focuses on the workforce and their individual needs respectively, companies are required to listen actively, create a healthy work environment, provide benefits that correspond with personal needs, and engage their employees pro-actively in order to make their business thrive and convince employees that their business is the best place to work [100]. Workplace health promotion is a promising way of achieving and strengthening identification [101]. WPHP comprises aspects beyond workplace safety and health promotion—namely, personnel management and staff development [102]. A recent survey produced evidence that WPHP measures are one of the top three priorities amongst employees of all ages [103].

Organizations can affect health in a variety of ways—for instance, through material, behavioral, and physical mechanisms [104]. In the process of WPHP, work context factors are of relevance [105,106] and have been found to influence employee health and well-being [107]. These, for instance, concern ergonomics (i.e., the extent to which an appropriate amount of movement and posture is possible in the workplace); physical demands (i.e., the amount of physical effort required to perform the job); work conditions (i.e., pleasant physical conditions in the workplace, including temperature, low safety risks, and low noise levels); and equipment (i.e., the degree to which a job requires a variety of equipment). Investments in WPHP have been found to pay off in the long run, resulting in increasing productivity and lower turnover rates [16,48,108,109,110].

## 3. Empirical Study

### 3.1. Study Purpose

The focus of the empirical study concerned healthy leadership and employee well-being in the digital workplace. Based on the previously outlined generic organizational health framework by Bauer and Jenny [52,53] and the strategies and factors to enhance organizational health and respective WPHP measures introduced before, the empirical investigation seeks to investigate how individuals cope with stress. In addition, we seek to explore whether their supervisors (leaders) lead by example by following a healthy lifestyle, while also inquiring the existence and support generated by the organization (in the form of WPHP).

Besides individual strategies of dealing with this dichotomous approach to health, the empirical study aims to determine whether employees feel that companies increasingly step up and recognize the workplace as a place where individual health can be enhanced. As such, it is proposed that organizations draw from their health promotive capacities.

In detail, the study examines, from an employees’ perspective, (1) whether employees themselves perceive to be in charge of their health and (2) the strategies they utilize to manage the often contradictory requirements of their private and professional lives (work-family-balance), (3) whether companies have already established a healthy leadership and whether their leaders model healthy lifestyles, as well as (4) whether employers have already implemented WPHP measures. If this is the case, we also seek to uncover how employees evaluate the respective WPHP measures. Hence, the relevance of individual as well as organizational strategies to optimize individual and organizational health is scrutinized.

### 3.2. Method

Semi-structured qualitative interviews with 40 (full and part-time) employees from two European countries (Austria and Germany) were conducted over the course of two months. We used purposeful sampling in order to recruit interview partners that could provide in-depth and detailed information on healthy leadership and workplace health promotion [111]. For this purpose, we identified a set of qualifying criteria each participant had to meet to be considered for the study. The interview partners were selected if they fulfilled the following criteria: they (a) were employed either full-time or part-time, (b) have been with their employer for six months at the minimum, and (c) indicated to be interested in their health. Moreover, we ensured to recruit an equal amount of females and males, as well as representatives from different industries and management levels (leadership vs. non-leadership positions). In order to recruit interview partners, we turned to a German and Austrian corporate database. Each interview was conducted face-to-face and fully transcribed afterwards; the transcripts served as the basis for our qualitative content analysis. All interview transcripts were analyzed using QCAmap. The respondents (female: 27; male: 13; age range: 22 to 57 years) worked in a variety of industries, such as tourism, banking, IT, and security. About half of the participants (*n* = 19) did not have a management position. The average interview time was 35 min.

A qualitative research approach was chosen, as, instead of confirming existing research findings, the main focus of the chosen method is to get new insights [111]. The interview guideline consisted of four broad categories: (1) Importance and Definition of Health, (2) Health in the Workplace, (3) Workplace Health Promotion and Healthy Leadership, and (4) Work-Life Balance. Sample questions for each category can be found in Table A1 in Appendix A.

Semi-structured interviews were used in order to give the interview partners freedom to elaborate on selected aspects [112,113,114]. Interview material was analyzed via a content analysis, as proposed by Mayring [115,116]. According to Mayring ([115], p. 114), the advantage of a qualitative content analysis is “systematically analyzing texts by processing the material step by step using category systems that have been developed based on theory”. In the process of the analysis, material is divided into units and summarized to answer the previously introduced research questions.

The fully transcribed interviews were saved in text format. All interviews were uploaded to QCAmap and analyzed using both a deductive and inductive approach. The analysis was deductive in that it tried to find support for the previously introduced topics, investigating the extent to which previous research findings are supported. We did so by assigning interview passages to pre-defined categories based on the literature—the categories were also reflected in our interview guideline. The analysis was, however, also inductive, as we tried to identify aspects that have not been thematized by previous research to date. In addition to inductive/deductive coding, we also were able to report the frequency of elements. A summary of the most important themes together with exemplary quotes can be found in Appendix A (see Table A2).

The content analysis was conducted by two independent coders. The individual coders were trained by the authors on how to code variables along the specific categories. The codebook was refined on several occasions to account for the additional insights that emerged during the training process. When coders had reached acceptable levels of reliability [117], they coded one-third of the interviews. Based on this sample, Krippendorff’s alpha was calculated for each coding category to ensure acceptable levels of intercoder reliability [117] and ranged from 0.82 to 1.0. All discrepancies were discussed and resolved before the final analysis was conducted.

## 4. Results

### 4.1. Individual vs. Organizational Responsibility for Health

In a first step, we asked participants about the relevance they attribute to their health and, also, who they think should be responsible for their health. To the largest part, individuals attested that they were in charge of their health. Exemplary statements read as follows: “I am responsible for my health, I would say” (male, 28, IT industry), “I am solely responsible for my health” (female, 23, hospitality industry), and “I pay a lot of attention to my health and try to improve it on multiple occasions” (male, 28, IT industry). While “everyone is responsible for their own health” (40, male, sales director), there was also agreement that individuals are able to improve their health and well-being more readily, “because these days, you have many opportunities to take care of yourself” (female, 23, education). One respondent, nonetheless, remarked that his health was subject to environmental influences, including the workplace (male, 28, engineering).

When inquired as to how participants took care of themselves at work, several measures were listed, such as “getting up and moving around in-between work. Taking the steps instead of the elevator” (male, 54, public service), “take enough breaks, eat on a regular basis” (female, 24, education), or “go outside for a bit” (female, 24, education). One interviewee even mentioned that he learned how to “sit in front of the computer for 8 h, or how can I practically strengthen my eyes” (male, 54, public service). In one company, exercise was not an individual but a team effort: For example, the employees said “Let’s go up a few floors” or “Let’s take some exercise. That’s convenient” (female, 31, public service).

### 4.2. Work–Life Balance

Strategies to establish a healthy work–life balance entailed, for example, flexible working hours (male, 28, IT industry). In the context of work flexibilization, the largest proportion of employees indicated that separating their private from their professional lives was quite challenging, opting for an *integration*. One employee stated that “with the modern technologies, it is very hard to separate work from home” (male, 47, service industry). Respondents attested that work presented a central component of their private lives, forcing them to integrate work-related aspects into their homes on a regular basis. Conditioned by the omnipresence and broadening scope of new technologies, the two spheres are “more and more connected because with these new tools, like mobiles, where you get your emails, and it gets more complicated to really disconnect” (male, 40, sales director). This also held true for an employee from the banking industry, who observed his own inability to “escape it. Your work will influence your private life and your family will feel that you are not content and preoccupied with job-related matters” (male, 53, banking).

One interviewee described it as a “balancing act, since the company is always very present in the private sphere. One does additional tasks. But it is exactly the other way around as well: it is possible to organize and do private things in the workplace” (male, 47, engineer), while one extreme case even forced an employee to attend to job-related matters during his honeymoon, attesting to his inability “to separate the two areas” (male, 28, engineering); another employee held a more relaxed attitude towards the subject matter: “Of course, it happens that I go for a drink with my work colleagues after work and that we talk about work. But I also think, yes, that’s just part of it, because that’s everyday life, work is part of your everyday life and you talk about it, too.” (male, 22, IT).

However, not all employees are content with integrating their professional lives into their private lives; instead, they decide to take pro-active action to ensure that the two spheres do not mix. This is commonly referred to as *adaptation*. Since “it is very important for me to separate these two” (male, 24, lawyer), individuals have to take action themselves to guarantee a break from work. While individuals can, allegedly, “at a certain level, […] adjust your life around the company” (male, 40, sales director), for others, it is not as easy. One interviewee, who is a sales manager, remarked: “It was quite difficult to draw a line in the beginning, because you just go into the stores in a different way. So that’s certainly the problem, that somehow you always know that you are in a market where an employee of yours works. But you learn to live with it” (male, 47, sales director). Another employee shared: “Until January, I actually worked from Monday to Friday, and my life has been sacrificed to the employer—no let’s say my life belonged to my employer. And now, through teleworking, I have 2 days a week when I can do things for myself, when I have time for the family and that’s very important for me” (male, 54, public service).

In some instances, a *separation* of the two areas is possible. In these instances, even employers ensured employees’ ability to disconnect from work, e.g., by not grating them the possibility to access work-related information from home. A female respondent expressed appreciation for this regulation, claiming that “you leave the office and only think about work when you come back the next morning” (female, 39, IT industry). This, however, did not constitute the norm, as others were forced to take matters into their own hands. For example, one employee postulated in order for his work not to interfere with family time: “I got myself a private cell phone” (male, 40, sales director). Another said: “I try not to work at home” (female, 24, education). This notion is shared by another interviewee: “At first, I couldn’t keep my work and private life separate, because I was doing a lot of work at home. But now I know, that if I go to the office, I do my job and if I go home, I don’t do a lot of work at home, so it helps me, to keep it separate” (female, 39, IT industry).

### 4.3. Healthy Leadership

Mixed results were reported with regard to the extent organizations in general or managers in particular expressed concerns for employee health. While some respondents claimed that health was not an organizational issue, announcing that “in my company, health doesn’t play any role” (female, 28, service industry), or “my employer does not play any role in my health” (42, female, public service), other employees reported more positive experiences. In line with the observation that “my environment can also influence my health” (male, 28, engineering), one female respondent, who works in education, stated: “At my workplace, it is great. We have an own department that deals with health promotion and management. And there are many offers I can choose from, provided I take advantage of them” (female, 24, education).

At times, managers take a hands-on approach to health as well. One employee claimed: “My boss is one of those people, who puts a carafe of water on my table when I cannot get any. So he most certainly takes care of his employees” (female, 23, manufacturing). Another respondent stated that their work canteen offered healthy food, and “our boss often joins us for a healthy meal” (female, 30, manufacturing). In other cases, the whole department attends gym classes together, and that also includes the top management (male, 28, sales). One interviewee shared a very positive example: “My superior offers weekly appointments where you can come to her with concerns, but you can also talk with her about stress and she helps us when she recognizes that we can’t deal with the situation ourselves” (female, 31, public service). Through healthy leadership, managers can positively shape responses to organizational offerings and lead through their own behaviors, thus functioning as role models.

### 4.4. Workplace Health Promotion (WPHP)

Albeit both employees’ concern for their health and calls for employers to take responsibility for individual health are on the rise, notions of how health in the workplace can be improved are quite diverse (male, 28, engineering). Sixty percent of all respondents indicated to have benefitted from corporate initiatives and showed appreciation for what they had been offered so far. Corporate WPHP measures, for instance, involved the provision of free fruits and vegetables (female, 22, hospitality), free health checks and vaccinations (female, 24, education), comfortable chairs and a pleasant office atmosphere (female, 39, IT industry), and gym classes/memberships either free of charge or at a reduced fee (female, 24, education).

When inquired about the relevance of WPHP measures related to a work–life balance, employees did not fail to express the importance of separating their private lives from their work lives. While respondents claimed it was definitely easier to keep the two areas apart for some industries (male, 32, hospitality; female, 42, service industry), personal efforts alone would not suffice but need to be complemented by corporate measures (male, 40, sales director). While employers only stepped up in selected cases, ensuring that e.g., employees were unable to access the work content from home (female, 39, IT industry), or even offering an employee some additional flexibility after he had become a parent (e.g., extra parental leave or home office time) (male, 28, engineering), similar corporate steps were also reported by a service industry worker, who reckoned: “They give you extra vacation up to two days a year so you can do good social things like helping in the kindergarten. They also give you an extra day for your birthday which you can use flexibly—meaning you have plus/minus a week to take it” (male, 47, service industry). An alternative would be appointing a deputy in chief, who then attends to matters (male, 53, banking). These incentives, however, do not present the norm yet. Overall, employees feel like there is room for improvement, stating that they would appreciate small steps, such as the opportunity of telecommuting (male, 28, engineering).

## 5. Discussion and Implications

Our study showed that employees attach a great importance to their health and confirmed that present-day conceptualizations of organizational health have to take shifting priorities on part of employees into account (see also, References [118,119]). These changes also correspond to the changes brought about by Work Culture 4.0 [120] and Work 4.0, respectively [121], which are not only conditioned but also enabled by digital technology. Our research again confirmed that ICT are a double-edged sword (see also, Reference [31]). Against the background of an increased workload and rising instances of workplace stress [5,6], our study showed that organizations have started to invest in workplace health promotion.

The study further confirmed that employers have begun to recognize the potential of the workplace as an environment, where individual health can be enhanced [81,93]. Several employees reported that the first steps towards a healthy leadership have already been taken in their company. As part of healthy leadership [23], managers’ responsibility and capability in advancing organizational health by leading by example were confirmed by our study. Corporations cannot only encourage individuals to take a proactive role in their health, but through healthy leadership, they can also positively shape responses to their offerings and lead through their own behaviors, thus functioning as role models [90]. Several participants acknowledged that employers or executives in a leading position take on a role model function, as proposed by Morgenroth et al. [91].

Likewise, our study validated that social relations with other team members or employees hold a health-enhancing capacity. Referring to social capital theory, strong support from or social relations with leaders and other peer group members can increase individuals’ feelings of support and empowerment [122]. Thereby, social support can take different forms, such as instrumental support (i.e., receiving help), emotional support, informational support (i.e., receiving advice and guidance), and appraisal support (i.e., receiving feedback [123]). Social support from within the organization thus presents an effective means to help buffer the negative effects associated with workplace stress [124]. As such, it becomes a valuable source to increase job performance, further fostering the development of a positive organizational climate [125,126].

As Bauer and Jenny [53,54] rightfully observed, the workplace as an environment, and individual health are reciprocally linked, as employee health is influenced by the organizational context, which, in turn, is influenced by employee health. Linking our study findings to the generic organizational health framework introduced by Bauer and Jenny [53,54], our results substantiate previous claims that organizations must take a holistic approach towards health, offering initiatives that involve employees, managers, and processes (work and social), as well as the organizational environment altogether. Organizations should implement health-enhancing measures for employees, e.g., by prioritizing employee health and well-being and making it an integral part of their supportive organizational culture and supportive leadership [34]. Although our study showed that ergonomic measures that concern the workplace infrastructure, layout and design, agreeable work conditions and appropriate equipment are very important and appreciated by employees, it is even more important to help employees achieve a good work–family balance. Employers should thrive to reduce the burden posed on employees by redesigning work tasks and implementing more flexible solutions. The increasing workload and corresponding developments (e.g., burnout, stress, etc. [13]) compel organizations to not only prioritize their profits but, also, their employees’ well-being and health. These calls are also in line with the United Nation’s SDGs, which emphasize the need for organizations to invest in decent working conditions and have already been thematized by selected workplace theories. For instance, the quality of work life (QWL [127]) is used to describe employees’ broader experiences of the workplace environment. It requires organizations to recognize that employees do have a life outside of work, further evoking them to optimize working conditions and contextual factors to improve employees’ experiences of the workplace [128].

One striking result brought about by the present survey, however, was that one solution does not fit the requirements of an ever more diverse workforce; instead, individuals’ needs and preferences have to be accounted for. As the workplace is undergoing some rapid changes, employees’ motives are shifting too, which might be conditioned by the broader set of skills required in the digital workplace [129]. Hence, it is important to also involve employees in these far-reaching change processes [130]. In order for employers to tap into the pulse of time and thematize changing employee expectations in the workplace, employee concerns need to be determined in advance; if taken up and implemented in the form of comprehensive organizational measures, both employees and employers can have a mutually beneficial relationship in a supportive work environment [131].

This also suggests that employees’ preferences and motives need to be evaluated against the background of changing lifestyles and values [119]. However. not only employees’ wishes are undergoing some changes; likewise, organizational notions of human resources are subject to change. As already mentioned above, the “war for talent” conditioned a paradigm shift, emphasizing the need to attract qualified employees, develop them, and keep them in the company [32]. As good human resources are difficult to replace, they represent one of the rarest and most valuable resources of an organization [35] and are part of the organization’s movement towards sustainability [33,36].

Based on our study findings, and in line with developments witnessed during COVID-19, flexibility and the option of remote work have been found to benefit employees tremendously in combining their private and personal lives. Amidst the turmoil brought about by the health crisis—and the consecutive labor disruptions—organizations have started to implement solutions that do not require employees to work in physical proximity. According to PwC [132], 83% of the US companies surveyed claim to have been successful in implementing remote work. Yet, employers are not convinced that remote work will present a permanent solution, as, in order to keep up a strong organizational culture, employees are expected to be present in the physical workplace. Nonetheless, remote work and hybrid workplace solutions are expected to be a permanent scenario in the future [133].

Right now, as our study with employees has demonstrated, employer concern for employee health and well-being is an add-on rather than a given; and while corporate efforts are slowly surfacing—and will presumably be intensified in response to COVID-19—steps towards a truly supportive organizational culture that is based on an understanding of employees’ needs have to be taken more proactively. While notions of organizational health are subject to environmental conditions and, as such, likely to change [134], organizational health efforts need to be made a part of everyday organizational practices [135,136]. Our study confirms that employers have started to recognize the potential of the workplace as an environment, where individual health can be improved sustainably. Some have even tapped into the concept of a healthy leadership, recognizing the possibility of mangers to “lead by example”. The role of managers in ensuring individual health becomes even more crucial if organizational health measures are absent, as managers’ commitment can positively effect employee satisfaction and engagement [34]. This supports the notion that, as leaders, they can serve as a role model [90], motivating employees to follow their example [92].

If organizations attempt to implement a “culture of health and well-being”, which is supported by “the institutional, social and physical environment” [137], a more profound understanding of employees and their individual life situations is needed. Yet, results show that only a few companies have already implemented WPHP measures, if only to a certain degree. At present, companies are still more likely to delegate responsibility for employee health and well-being to their staff, having not fully realized the potential of healthy leadership, which puts individuals first. Overall, there is a great potential to increase WPHP measures on the employer side, through both healthy leadership and supporting WPHP measures.

## 6. Conclusions 

Our study provided useful insights into employees’ perceptions of who is responsible for their health, as well as insights into healthy leadership tendencies and WPHP measures, which have already been established by companies in Germany and Austria. Yet, there are several limitations to the study presented in this paper, which offer directions for future research.

Our study looked at employees’ perspectives only; therefore, it would also be of interest to incorporate the employers’ perspectives. It would be also interesting to extend the study to other countries and to compare the establishment of healthy leadership and WPHP measures in different countries. For example, a recent industry survey showed that most U.S. American companies claim to only invest in very basic health strategies, meaning they only offer healthcare benefits or health initiatives but do not really pursue a healthy leadership approach [10].

In addition, future researchers could take possible moderators into account, such as cultural differences. The GLOBE researchers House et al. [138] identified several leadership styles in different countries, and so, it would be interesting to analyze in which country a healthy leadership is already being used or also expected by employees. Likewise, it would be fruitful to uncover how healthy leadership can be combined with other leadership types, such as supportive leadership [139], virtuous leadership [34], ethical leadership [140], or relation-oriented or transformational leadership qualities [90]. The cultural dimension of individualism and collectivism [141] could also be an important moderator for employees’ preferences regarding who assumes responsibility for their health. Besides cultural differences, individual differences could influence employee preferences and expectations regarding individual vs. organizational health efforts (e.g., health status, tenure with the company, internal locus of control, or personality traits such as neuroticism or a proactive personality; additionally, see Reference [4] for a discussion on the role of psychological capital in mitigating occupational stress).

Our study was conducted prior to the COVID-19 pandemic. As the pandemic is likely to increase economic and job uncertainty and, with it, has detrimental effects on employees’ psychological well-being [142,143], another direction for future research would be to replicate our study once again in the present in order to analyze whether and how COVID-19 has impacted the healthy leadership and WPHP measures; when doing so, it would be worthwhile to also scrutinize employees’ and employers’ perceptions of the healthy organization.

For our study, we chose a qualitative approach. Future researchers might want to use additional methods, e.g., it would be interesting to conduct a quantitative survey or expert interviews with executives from different companies.

We asked participants about their perceptions of already established WPHP measures and also inquired desired measures; however, the results could not be explored in detail in this paper, as it would exceed the scope of our paper. A selected set of recommended measures, however, is reported in Table A1 in Appendix A.

Future researchers might also want to focus on sociodemographic factors, especially those that influence individuals’ work–life balance such as gender, children in the household, or relationship status.

## Data Availability

Interview data is available upon request.

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
