# Peer review of "Healthy Leadership and Workplace Health Promotion as a Pre-Requisite for Organizational Health"

_ijerph, 2021, doi:10.3390/ijerph18179260_

Round 1
Reviewer 1 Report
Comments for the Author(s)
This qualitative study explores the extent to which: employees perceive the locus of responsibility for their health (self vs. employer), employee use strategies for work-family balance, companies have begun to implement health-focused leadership, and workplace health promotion measures have been implemented by organizations. While there is much to like about this paper, three major issues that prevent the paper from being published in its current form. Below I explain my major concerns along with suggestions for improvement. I also point out a few minor issues that should be considered and/or corrected.
Major issues:
1. Paper Framing and Contribution.
Although the topics of workplace health promotion and health-focused leadership are of potential interest to both health researchers and practitioners, the present study, as currently framed and presented, makes a relative modest contribution to our understanding of these important issues. I think the overall interest and contribution of the paper could be effectively enhanced by reframing the paper and by clarifying the methods, results, and leadership construct (see points 2 & 3 below).
On p. 2 at the end of the introduction, you clearly state the aims of the paper. However, I found myself asking why we should care about these aim. For example, on p. 2 and again on p. 3, you note that you are focusing on who is more responsible for employee health, the employee or the employer, and suggest that you will attempt to fill this gap in our knowledge. But why should this gap be filled? Why should we care about this question?
I suggest that you do a better job of framing the paper and highlighting its potential contribution in the introduction so that by the time we reach the study aims at the end of the introduction, we know exactly why these aims matter. One specific suggestion for framing the paper comes from something you mention in the discussion section on p. 9. You mention the need from a human resource management perspective to invest in competitiveness and skilled employees in order to create an inimitable and rare competitive advantage. This concept could be expanded and moved to the introduction to help frame your study and explain why we should care about the study’s aims and questions.
2. Description of Methods and Reporting of Results.
You could also further clarify the study’s contributions by more clearly describing the study’s methods and more clearly presenting the study’s results. Specifically, providing a detailed explanation of how the QCAmap tool facilitates the analysis of the structured interviews would be very helpful. In addition, a clearer presentation of the results would be useful. I suggest that you consider adding a summary table organized by key themes and associated illustrative quotes. This would help the reader to see the big picture of what you found from your analyses and how the findings contribute to our knowledge.
3. Clarification of the Healthy Leadership Construct.
I had some concerns regarding the healthy leadership construct. I understand that the term comes primarily from the Hansel and Kaz paper, which I cannot read because I do not read German. However, I would note that in English “healthy” can have a connotation of simply “good” or “functional” or “beneficial.” At a glance therefore, “healthy leadership” could be interpreted simply as “beneficial leadership.” I would prefer the term “health-focused leadership” (e.g., Boehm & Baumgärtner, 2014) or “health-specific leadership” (e.g., Gurt et al., 2011), which both appear to be more established constructs than “healthy leadership.” At a minimum, I think you should review these health-focused leadership constructs, differentiating them from one another and clearly identifying the specific health-focused leadership construct used in your study.
Minor issues and points of concern:
1. In both the abstract and first sentence of the introduction, you state “stress constitutes a concerning organizational trend.” But I’m not sure that stress can be a trend. Increasing stress can be a trend. Higher levels of stress could be a trend. Please clarify this point.
2. In the introduction you use the abbreviation WPHP before you introduce the full term and abbreviation. Please use the full term in the abstract.
3. You introduce the concept of “healthy leadership” (or whatever you term you might choose to use instead) in the introduction, but you do not explain it. A very brief definition/overview here would be helpful.
4. The first sentence of the results section on p. 6 is unclear to me. Please clarify your meaning.
5. Regarding the preference for individual vs. organizational health efforts (section 4.1, p. 6), it could be worth mentioning some potential moderators of this preference. Could this preference be affected by individual differences (e.g., proactive personality or internal locus of control) or by cultural differences (e.g., individualistic vs. collectivist cultures)? Germany and the U.S., for example, are both highly individualistic cultures, which could shape the perspectives reflected in your results. Would these perspective be different in a highly collectivist culture?
6. The first sentence on p. 8, “need to complemented” should probable be “Need to be complemented.”
Thank you for the opportunity to read your paper and make these suggestions. I wish you the best as you move forward with the paper!
References
Boehm SA, Baumgärtner MK (2014) Health-focused leadership: prevention and intervention as enablers of followers’ health and performance. Paper presented at the 74th annual meeting of the Academy of Management, Philadelphia, 1–5 Aug 2014
Gurt J, Schwennen C, Elke G (2011) Health-specific leadership: is there an association between leader consideration for the health of employees and their strain and well-being? Work Stress 25(2):108–127
Author Response
Thank you for your thorough review.
We have addressed your suggestions in our revision - for a detailed overview, please consult the attached file.
Thank you once again for giving us the opportunity to revise our manuscript.
Best regards,
Isabell Koinig and Sandra Diehl

Reviewer 2 Report
Dear authors,
I have inserted several comments along the paper.

Author Response

(The authors gave the same response as above.)

Reviewer 3 Report
1. Emphisize how the paper closes the literature gaps in the introduction, and innovative points in the discussion section. The originality of the paper is questionable so make it more clear why the study is innovative and significant.
2. The overall logic and coherence of the paper is of a good quality, but theoretical review should be enhanced with more relevant papers. The article reviews non recent literature.
You should review new articles published in 2020 and 2021. Many significant studies were published in last two years so the paper should cite articles from 2020, 2021.
For example you can include these, and some other from 2020, 2021:
Obrenovic, B., Jianguo, D., Khudaykulov, A., & Khan, M. A. S. (2020). Work-family conflict impact on psychological safety and psychological well-being: A job performance model. Frontiers in psychology, 11, 475. Godinic, D., Obrenovic, B., & Khudaykulov, A. (2020). Effects of Economic Uncertainty on Mental Health in the COVID-19 Pandemic Context: Social Identity Disturbance, Job Uncertainty and Psychological Well-Being Model. International Journal of Innovation and Economic Development, 6(1), 61-74. Guberina, T. & Wang, A.M. (2021). Entrepreneurial Leadership Impact on Job security and Psychological Well-being during the COVID-19 Pandemic: A conceptual review. International Journal of Innovation and Economic Development, 6(6), 7-18.3. Very little information on how the data was collected. Has the study been done during COVID-19? Please provide more information to the methods section, on data collection, sampling method, more rich information on the sample itself, and demographic info on the companies.
4.Results should be more analytical with percentages supplied and avoid terms such as "around half". I would also like to see a systematical arrangement of the results, even though it is a qualitative study. Please structure the data , and make a table which reports some results systematically.
5. What are the suggestions for future studies? The current suggestions are not Introduce some COVID-19 specific suggestions, especially because in the end you mentioned COVID.
Author Response

(The authors gave the same response as above.)

Reviewer 4 Report
Dear authors, it is a pleasure to read your interesting manuscript.
Here are some points to improve it:
- Question born after the complete reading of your paper: is "healthy leadership" the actual main focus of your research? Or is it only a peculiar aspect? For example, you do not cite it when you state the study aims in the introduction.
- Abstract: it may be useful to specify, at the first time, what "WPHP" stands for.
- Page 2 line 52: please clarify the sentence. Same at line 55 for the citation.
- Page 2 line 61: please specify what is meant by "flexible working solutions".
- In the literature review you write: "To date, little research has been done on who is more responsible for employee health- the employee or the employer. This is a research gap that our study intends to fill". Do you really find it important? Is it not enough to say that both workers and organisations are responsible for people's health?
- "In recent years, the term occupational health has been substituted by organizational health [43]". Is it a sostitution or instead is it a change of approach to make the topic addression more comprehensive?
- Please, specify more details about the content analysis you performed, in particular by adding info on the Mayring's approach and the functioning of QCAMap. For example, how did you select the content to be reported?
- You report the aims of the study and its results. On the other side, it not clear what you asked, specifically, to employees. Could you provide more info about this point?
- Discussion and implications are fine, but maybe to provide some lists, or point-by-point "recap" may help the reader to be oriented in the paper
- The last pagraph states about limitations and further research. Maybe a conclusion with, again, a recap may "overwrite" in the reader the limits underlined before :).
Congratulations on your manuscript and best wishes for the further steps!
Best regards
Author Response

(The authors gave the same response as above.)

Reviewer 5 Report
Healthy leadership and WPHP as prerequisite for organisational health
The paper sets out to explore how employees perceive their work environment about health and how their employers engage in healthy leadership. The paper has a well written literature section but needs to be improved regarding the aim, the methodology, and the presentation of results. Much would be gained by a clearly stated aim and a clearly selected level and unit of analysis.
Major issues
The aim of the study needs to be reformulated. State the same aim on all places in the paper. My recommendation is that you use one aim for the paper focusing on employees and their experiences/strategies, e.g., “the purpose of the study is to explore how employees perceive their responsibility for their health and which strategies they use to establish work-life balance.” You do not need all the secondary aims and research questions described. Instead focus on one level and one unit of analysis. Also, regarding the aim, some of the listed aims are better pursued with a quantitative design, e.g., “investigate to the extent to which…”.
I recommend that you focus on the employee perspective and their experiences of workplace health promotion. If you are seeking to investigate employer’s perspective they should be involved in the study, i.e., you would need to make interviews with employers. This is a major flaw in the present presentation. If you focus on employees, this flaw will be remedied.
The methods section needs to be further developed. Please present some sample questions from the interview guide or submit the full interview guide as an additional file or in an addendum.
You need to add a section describing the data analysis in more detail. How were data analysed? By whom? Which steps were taken in the analysis? This is a major flaw and needs to be elaborated. This is especially important for qualitative research. Remember, other researchers should be able to replicate your study based on the description in the methods section. You may also look at the SPICE-framework for qualitative studies for inspiration.
Other important methodological questions that need to be addressed:
- What is the level of analysis in the study?
- What is the unit of analysis in the study?
- How were interviewees selected? Randomly? Strategic sampling? Convenience sampling?
- What were your relation to the interviewees?
In the results section, were the categories (subheadings)chosen a priori or are they a result of the data analysis?
The first subheading does not match the content of the paragraph. A better subheading would be “individual responsibility for health”.
Under the subheading work life balance results regarding strategies for work-life balance is presented. However, there are only two such strategies: flexible working hours and disconnection from work platforms by the employer.
Minor issues
Abstract: State the aim of the study in the abstract (after line 10) and move the sentence “for this purpose…” (line 14) to line 10, after the aim.
The gap that appears on line 96 could be moved to the introduction since this is one of the main arguments for conducting the study.
Line 178: how is it a heuristic model of occupational health?
Line 301-304: this is an analytical statement which should be moved to the discussion section.
Line 409-413: this statement is not supported by the presented results. Please remove.
Author Response

(The authors gave the same response as above.)

Round 2
Reviewer 1 Report
The paper is much improved. I greatly appreciate the efforts of the authors in responding to my suggestions in their revision. I believe the paper now makes a contribution worthy of publication in IJERPH. Congratulations!
Author Response
Thank you very much for your positive feedback - your comments were essential in improving our manuscript!
Best regards,
Isabell and Sandra